# Perturbation biology links temporal protein changes to drug responses in a melanoma cell line

Elin Nyman[1,2,3,4☯]*, Richard R. Stein[2,5,6,7☯¤]*, Xiaohong Jing[8], Weiqing Wang[8], Benjamin Marks[1], Ioannis K. Zervantonakis[1], Anil Korkut[9], Nicholas P. Gauthier[1,2,7‡]*, Chris Sander[1,2,7‡]*

1 Department of Cell Biology, Harvard Medical School, Boston, MA 02115, U.S.A., 2 cBio Center, Department of Data Science, Dana-Farber Cancer Institute, Boston, MA 02215, U.S.A., 3 Department of Molecular and Medical Genetics, Oregon Health & Science University, Portland, OR 97239, U.S.A., 4 Department of Biomedical Engineering, Linköping University, Linköping 58185, Sweden, 5 Harvard School of Public Health, Boston, MA 02115, U.S.A., 6 Department of Systems Biology, Harvard Medical School, Boston, MA 02115, U.S.A., 7 Broad Institute of MIT and Harvard, Cambridge, MA 02142, U.S.A., 8 Memorial Sloan Kettering Cancer Center, New York, NY 10065 U.S.A., 9 University of Texas MD Anderson Cancer Center, Houston, TX 77030 U.S.A.

☯ These authors contributed equally to this work.
¤ Current address: Novartis Institutes for BioMedical Research, Basel 4056, Switzerland
‡Joint senior authors
* pertmel.a2058@gmail.com (EN); pertmel.a2058@gmail.com (RRS); pertmel.a2058@gmail.com (NPG); pertmel.a2058@gmail.com (CS)

**Data Availability Statement:** All data is available from https://gitlab.liu.se/eliny61/perturbation-biology-time-resolved.

## Abstract

Cancer cells have genetic alterations that often directly affect intracellular protein signaling processes allowing them to bypass control mechanisms for cell death, growth and division. Cancer drugs targeting these alterations often work initially, but resistance is common. Combinations of targeted drugs may overcome or prevent resistance, but their selection requires context-specific knowledge of signaling pathways including complex interactions such as feedback loops and crosstalk. To infer quantitative pathway models, we collected a rich dataset on a melanoma cell line: Following perturbation with 54 drug combinations, we measured 124 (phospho-)protein levels and phenotypic response (cell growth, apoptosis) in a time series from 10 minutes to 67 hours. From these data, we trained time-resolved mathematical models that capture molecular interactions and the coupling of molecular levels to cellular phenotype, which in turn reveal the main direct or indirect molecular responses to each drug. Systematic model simulations identified novel combinations of drugs predicted to reduce the survival of melanoma cells, with partial experimental verification. This particular application of perturbation biology demonstrates the potential impact of combining time-resolved data with modeling for the discovery of new combinations of cancer drugs.

## Author summary

Data-driven mathematical modeling of biological systems has enormous potential to understand and predict the interplay between molecular and phenotypic response to perturbation,

**Funding:** EN acknowledges support from the Swedish Research Council (Dnr: 2016-00244), RRS and CS from the National Resource for Network Biology (NRNB) within the U.S. National Institutes of Health (grant P41 GM103504). CS acknowledges additional support from the U.S. National Institutes of Health (Center for Cancer Systems Biology, grant U54 CA148967). The funders had no role in study design, data collection and analysis, decision to publish, or preparation of the manuscript.

**Competing interests:** The authors have declared that no competing interests exist.

and provides a rational approach to the nomination of therapy. In cancer, intense effort has focused on drugs that specifically target the machinery involved in tumor development, maintenance and response to therapy. Although many drugs have clinical efficacy in a fraction of patients, the response is rarely durable and patients often develop resistance within months. We believe that the robustness and complexity of living cells, as well as inaccurate assumptions about drug specificity in model systems, underlie the inadequacy of single-agent targeted therapy. In this work, we developed a framework to derive mathematical models of biological systems from molecular and phenotypic temporal responses, and test this framework on melanoma cells. These models are computationally executable and can be used to predict drug combinations likely to be effective at slowing growth or killing cancer cells. Our framework has several advantages: (1) drug specificity is learned, not assumed, during model training, (2) training on temporal (not static) response data improves predictive power, (3) data-driven dynamic models have the potential to accurately reflect a cellular system's behavior in a context-specific manner.

## Introduction

Targeted therapies are an important component of precision oncology as these agents—as opposed to standard chemotherapy—aim to counteract specific activating genetic or signaling pathway alterations and often have fewer side effects than conventional cytotoxic chemotherapy. Many targeted therapies have been approved by regulatory agencies for the treatment of various cancers [1]. However, initial response rates are generally not durable and tumors eventually develop resistance.

In melanoma, tumors with the common BRAF V600E/K gain-of-function mutation have been shown to have a remarkable response to drugs that specifically target the mutated protein kinase, such as the RAF inhibitor vemurafenib [2]. However, not all patients with a BRAF V600 mutation respond to targeted therapies, and the patients that do respond often develop resistance after only a few months. There are several known mechanisms of this resistance including primary resistance from loss-of-function mutations in PTEN that increase AKT signaling and reduce apoptosis, and CDK4 mutations and CyclinD1 amplification that promote cell cycle progression [3]. Mechanisms of acquired resistance include reactivation of the MEK/ERK pathway with new mutations in BRAF or NRAS and hyper-activation of receptor tyrosine kinases [3]. There is an urgent need for a more comprehensive understanding of resistant tumor cells in order to identify non-trivial therapeutic opportunities beyond targeting single genes.

We have previously developed a perturbation biology approach that simulates data-driven models to find combination vulnerabilities. These models are derived from measurements of the proteomic and phenotypic response of cancer cells *in vitro* to numerous drug combinations. For example, in a melanoma cell line, we predicted that a combination of a RAF or MEK inhibitor with a bromodomain inhibitor would be effective and synergistic and thereby reduce viability [4]. Such a combination (MEK with bromodomain inhibitor), has been subsequently proposed for a phase I/II clinical trial for small cell lung cancer and solid tumors with mutations in RAS [5]. Our approach has also been useful in identifying a synergistic drug combination (CDK4i with IGF1Ri) in dedifferentiated liposarcoma [6]. In these studies, we measured the protein response at a single time point and were thus not able to directly capture the transient cellular response to the drugs.

Several modeling frameworks to study cellular responses to perturbations over time have been proposed, for instance, ordinary differential equation models, dynamic Bayesian networks, and Boolean networks. Reaction-based models are assuming continuous dynamics and their kinetic parameters are inferred from data. Based on these fitted parameters they are executable and can be used to simulate the effect of unseen perturbations which can then be tested experimentally. They have been deployed to understand cellular responses to perturbations (e.g., [4, 7–11]). Dynamic Bayesian networks are stochastic and time-discrete and estimate probabilities of interactions based on data for the previous point in time, and have, for example, been used to study time-series observations in cancer [12]. Boolean networks are based on binary switches, and are discrete in time and state. However, this simplification of the underlying network, even though fuzzy logic variants allow for a softer discretization, is a rather strong one. Their benefit lies in the fact that less data is needed. Boolean networks have also been used to model cellular response data [13–15]. Only few of the above mentioned approaches qualify to be called large scale (i.e., including more than 100 states) [4, 7, 13]. However, none of these—in contrast to the work presented here—is based on time-series observations.

We hypothesize that short-term responses to therapeutic intervention, implemented by adjustments in signaling networks, already reflect the shifts in cellular processes ultimately implemented in cellular long-term memory via genetic changes. Typically these long-term adaptations are amino acid changing mutations ('missense mutations') or amplification/deletion of DNA fragments ('copy number changes'). It may therefore be possible to obtain evidence for the dominant resistance mechanisms in response to targeted intervention by observing the short term molecular signalling adaptations. In addition, time-series observations of response provide informative input to parameter inference for dynamic models, such as the one developed here, including the directionality, strength and sign of interactions [16, 17]. In order to study the short-term response to targeted therapies in melanoma, we produced time-resolved antibody-based measurements of protein and phospho-protein levels as A2058 cancer cells respond and adapt to drug perturbations. A2058 is resistant to both RAF and MEK inhibitors [18], and therefore represents melanoma patients that would fail one of the standard treatments. The obtained response profiles are used to infer interaction parameters of time-resolved mathematical models, which are selected for sparsity and for the ability to predict left-out data at reasonable accuracy. We simulated the consequences of all possible perturbations to these models when it comes to reduction of cell growth and induction of apoptosis. The results of these simulations were ranked by their predicted effectiveness. The most promising candidates, which are expected to reduce cell growth and increase apoptosis in melanoma, involve targeting the combination of IRS1 and EGFR. We experimentally tested the drug combination of NT157 (IRS1 inhibitor) with gefitinib (EGFR inhibitor) in A2058 cells, and find that (1) the drugs alone reduces cell growth substantially at high doses, and (2) the drugs in combination reduces cell growth further than any of the drugs alone.

In contrast to models of cell biological processes based on response data at a single time point (assumed to be steady state), deriving models from time-series data may allow one to better capture the dynamic contributions of molecular signaling processes to complex phenotypes such as cell proliferation and apoptosis. This systems biology paradigm of data-driven predictive dynamic models can be applied to other cancers, especially useful for tumors that are resistant to targeted therapies.

## Results

The goal of this data-driven systems biology study is to identify actionable cellular vulnerabilities and as well as to nominate novel vulnerabilities for future drug development. Conceptually,

we generated network models that link protein signaling to phenotype. These models are similar to static textbook pathway diagrams, but have the added benefit of being (1) highly specific for the system under study and (2) based on a well-defined mathematical formalism. In this study we produced dynamic (time-resolved) antibody-based measurements of protein and phospho-protein levels as the melanoma cell line A2058 adapts and responds to drugs (alone and in combination). These molecular and phenotypic data were used to train differential equations models that capture cellular dynamics. Using these models, we predicted and optimized the effects of untested molecular perturbations on cell growth and apoptosis (Fig 1).

## Experimental workflow and data

We characterized the temporal response of proteins, phosphorylation sites, cell growth and apoptosis, after application of drugs alone and in combination. A2058 cells were systematically perturbed with single drugs alone (both low and high doses) or in combination (low doses only), which resulted in 54 total drug conditions (Fig 2).

**Selection of drug concentrations.** For each drug, the concentration was chosen by literature review or experimental dose response measurements by western blot. The *low dose* was selected to reduce activity on a known target by 50%. We also used a *high dose*, which was defined as double the absolute concentration of the *low dose*.

**Molecular measurements.** Cells were collected at 8 time points after drug addition in logarithmic progression (10, 27 and 72 minutes and 3, 9, 24, 48, and 67 hours). The temporal proteomic response of the cells to the different conditions was derived using reverse phase protein array (RPPA) measurements in 124 total and phospho-protein levels [19] from minimally three biological replicates across two data sets. Antibodies were selected to broadly cover signaling pathways with known involvement in cancer (e.g., AKT, ERK, and JAK/STAT pathways).

**Phenotypic measurements.** We used live cell imaging to follow A2058 cells as they responded to drug treatment (Incucyte, Essen BioScience, Ann Arbor, MI, U.S.A.). We acquired GFP, mCherry, and phase images of all conditions every 3 hours for 72 hours following drug addition. Cell number was determined at each time point by counting using image segmentation software (Incucyte), which identified the transgenic H2B-mCherry fluoraphore. Apoptosis was determined by image segmentation and counting of the GFP channel, which measured a Caspase-3/7 fluorescent activity reagent (Essen BioScience). Data were sub-sampled to select time points that closely matched collection of the proteomic data (1, 3, 9, 24, 48, and 67 hours).

**Agreement of experimental results with literature.** As most of the selected drugs have well described signaling effects, i.e., kinase substrates whose activity is directly affected by the drugs (S1 Table), we asked whether each drug elicited the expected molecular changes. The drugs that target MEK, AKT, and JNK, resulted in the expected decrease of their known targets ERK1/2-pT202/T204, AKT-pS473/pT308, and CJUN-pS73, respectively (S2 Fig). RAF and mTOR inhibition did not result in a strong decrease in their direct targets (MEK1/2-pS217/221 and S6K-pT389, respectively), but they did significantly decrease signaling in further downstream nodes ERK1/2-pT202/T204 and S6-pS235/236. For the STAT3 and SRC inhibitors, the proteins' highest-ranked responders are not known to be downstream targets, but may be indirectly affected via multiple steps, signaling feedback or crosstalk.

The live-cell imaging data (see Materials and methods) revealed the bromodomain inhibitor CPI203 (BET inhibitor) to be the most potent drug, at the chosen concentrations, in terms of both decrease in cell growth and increase in apoptosis after three days of treatment (S1 Table, S1 Fig). These data are in agreement with results on the effect of this inhibitor class in

## 1. Perturb cells with drug combinations and measure cellular response

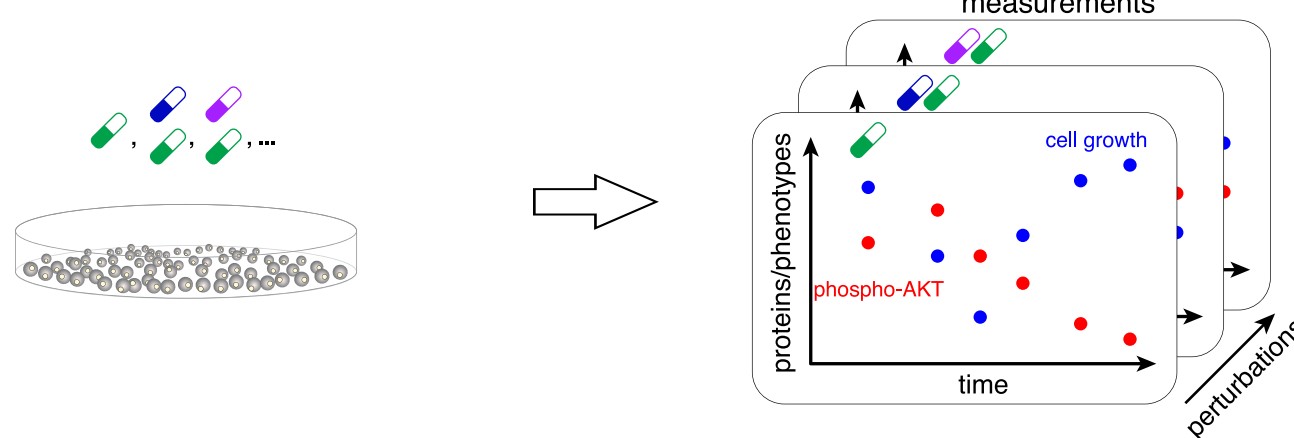

## 2. Fit ODE models to measurements

$$\frac{\mathrm{d}x_i^{\mu}(t)}{\mathrm{d}t} = \varepsilon_i \tanh\left(\sum_j w_{ij} x_j^{\mu}(t) + u_i^{\mu}(t)\right) - \alpha_i x_i(t)$$

interaction node perturbation

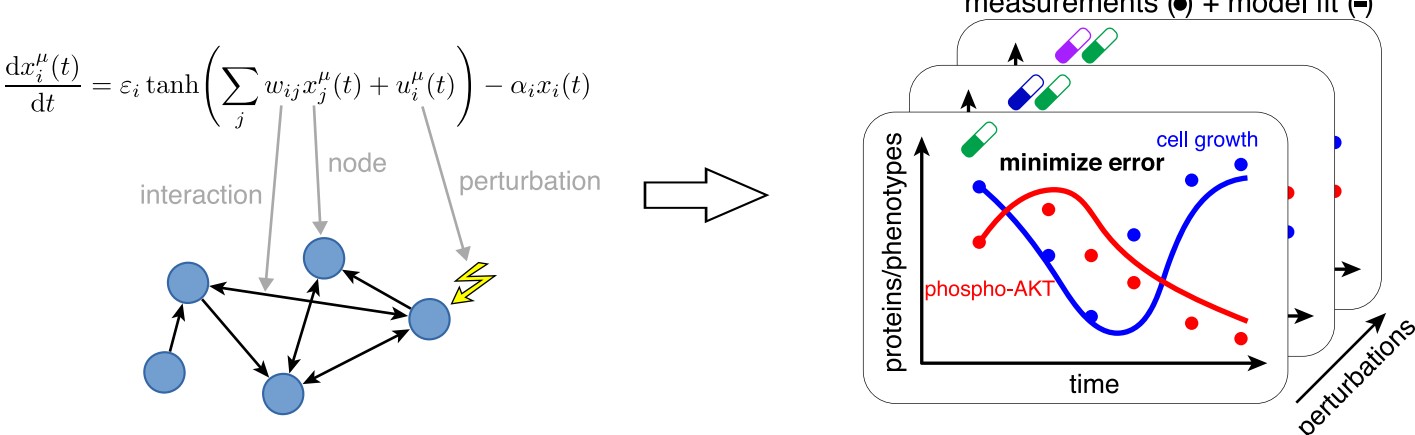

## 3. Predict effect of novel perturbations and search for combinations that reduce viability of cells

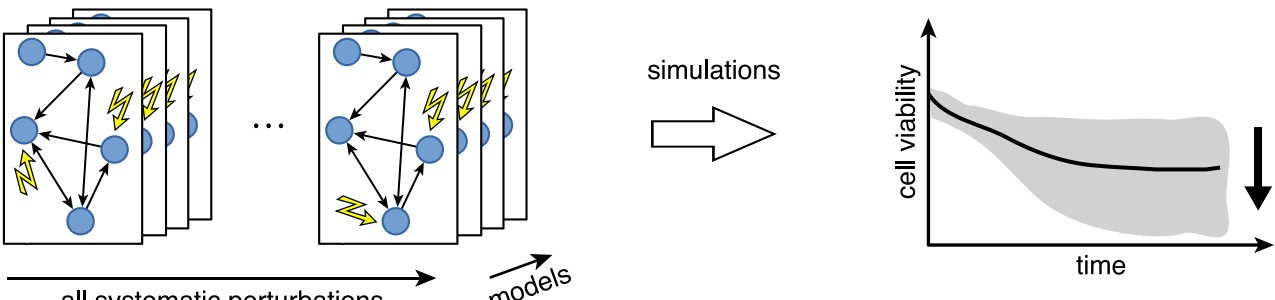

**Fig 1. Systematic experimental perturbation of cells leads to predictive network models.** We perturbed cells with combinations of targeted drugs and measured the time-resolved cellular response (step 1). These measurements were used as input to derive network models of the response to arbitrary combinatorial perturbations (step 2). Using these models, we identified drug combination targets that optimally reduce cell growth and increase apoptosis in a melanoma cell line (step 3).

## Dynamics of response to single and pairwise drug combinations

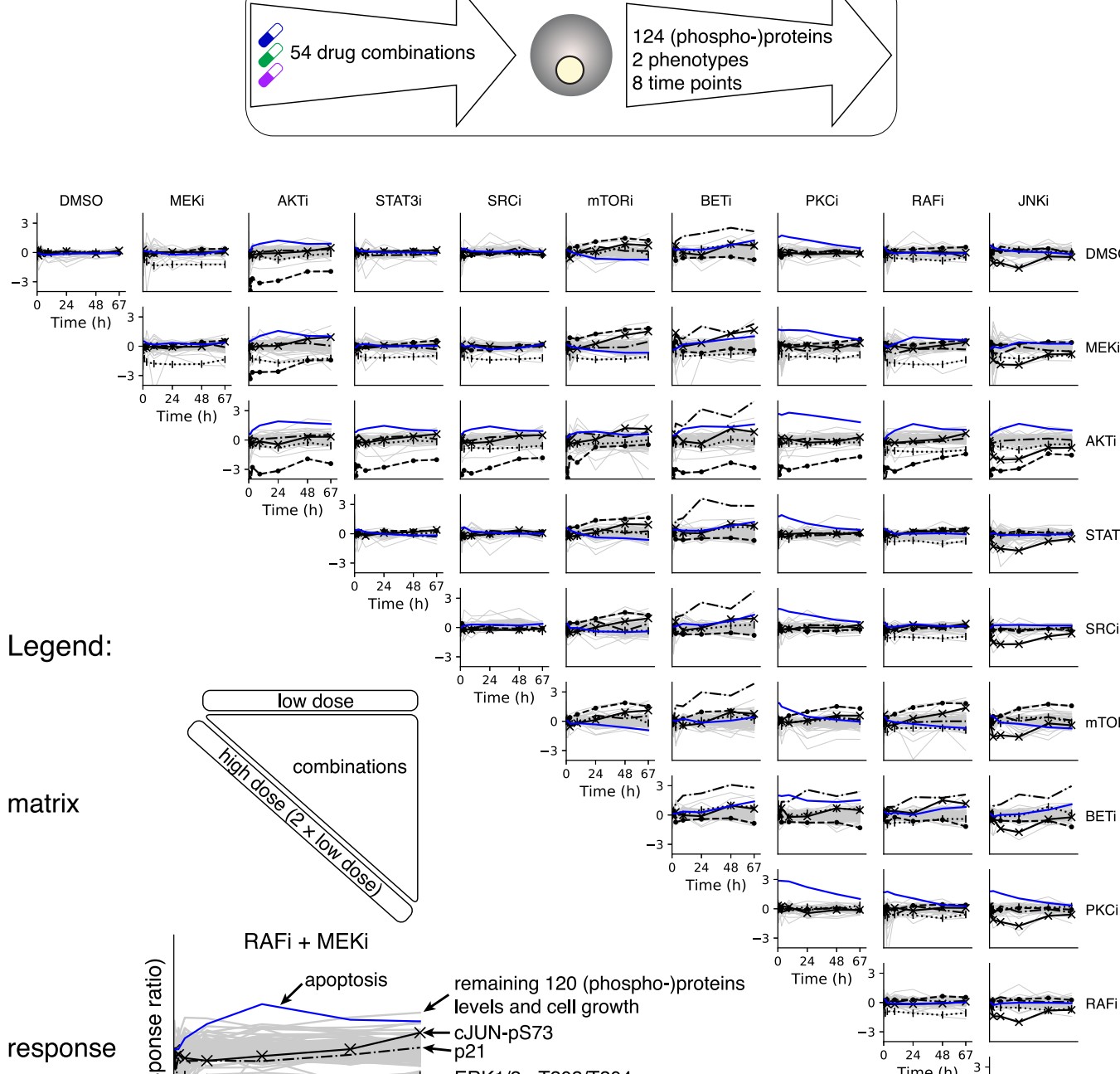

**Fig 2. Phenotypic and proteomic response to single and drug combinations.** The melanoma cell line A2058 was subjected to 54 drug combinations and the response of 124 (phospho-)proteins and two phenotypes (cell number and apoptosis) was measured at eight time points from 10 minutes to 67 hours. These data were used as input for model inference and prediction. The most informative data involved a subset of the proteins and phospho-proteins (black lines) that underwent the largest changes upon perturbation (AKT-pS473/474, ERK1/2-pT202/T204, cJUN-pS73, and p21) as well as apoptosis (blue line). The remaining 120 proteins and phospho-proteins together with cell counts had a less pronounced response to perturbation. Temporal response (vertical axis) is defined as $\log_2(x^{\text{perturbed}}(t)/x^{\text{unperturbed}}(t))$ where $x(t)$ are concentrations or counts as in Eqs 1 and 2 and DMSO is the unperturbed control.

the drug-resistant melanoma cell line SkMel-133, which also strongly impaired cell growth in response to the bromodomain inhibitor JQ1 in combination with MEK and ERK inhibitors in our earlier work [4].

## Network model construction and simulation

To analyze the dynamics of molecular and phenotypic response to drug perturbation and to predict the effect of unseen perturbations, we inferred and simulated network models from our rich experimental data (Fig 1). These data were divided into three disjoint subsets (top left of Fig 3). We first inferred the parameters of Eqs 2–4 on a subset of data (responses to single drugs, training dataset) by minimizing Eq 5 with added regularization term (Eq 6). The optimal regularization parameter $\lambda^*$ was then identified on a second subset of data (validation dataset) as that which minimized Eq 5 using the previously inferred model parameters. Finally, the model accuracy for the selected parameter set was estimated on a third dataset (test dataset). We then derived multiple models on the full dataset to predict the effect of unseen perturbations and rank them by their desired effect on cellular phenotype.

**Network models.**    To model the cellular processes, we used the previously proposed nonlinear multiple input–multiple output model [4, 20, 21]. This model has the ability to capture a wide range of biological and kinetic effects, and has been applied to diverse biological problems including to predict the effect of novel drug combinations [4, 6].

$$\frac{\mathrm{d}x_i^\mu(t)}{\mathrm{d}t} = \varepsilon_i \tanh\left(\sum_j w_{ij} x_j^\mu(t) + u_i^\mu(t)\right) - \alpha_i x_i^\mu(t).$$

(1)

The time-dependent variable $x_i^\mu$ describes the dynamics of node $i$ (in our context a protein level or measured phenotype) and $u_i^\mu$ the impact of a perturbation on node $i$ (e.g., the application of a single drugs or drug combinations) in experimental condition $\mu$. Here, $w_{ij}$ denotes the interaction between the nodes $i$ and $j$ (more specifically, molecules, phenotypes, drugs or processes). Intuitively, $w_{ij} > 0$ corresponds to activation of node i by node j, and $w_{ij} < 0$ to inhibition. The prefactor $\alpha_i$ quantifies the rate at which node $i$ returns to its initial state $x_i = 0$. The sigmoidal transfer function tanh is used to cap the reaction rates and to account for saturation and noise. Consequently, the prefactor $\varepsilon_i > 0$ determines the dynamic range of node $i$ as values of tanh are limited to the one-dimensional unit ball $[-1, 1]$. To obtain data-derived network models, we used Eq 1 and constrained its parameters to the proteomic and phenotypic measurements in 54 drug combination conditions and at 8 time points. Our full model contained 124 molecular and two phenotypic nodes. For a complete description of the model equations, see Materials and methods.

**Model selection and error estimation.**    Datasets were subdivided into training, validation and test sets (gray, green and blue boxes, respectively, in Fig 3 top left). The training sets contained all data from single-drug experiments; the validation and test sets were constructed such that (i) they contain all pairwise drug combinations and (ii) every drug has the same number of occurrences in both sets. Model parameter estimation was performed on the training dataset. We varied the regularization parameter $\lambda$ on a linear grid and trained 10 network models per grid value. As expected, higher values of $\lambda$ resulted in sparser networks with fewer interactions (Fig 3 top right, magenta) and higher error on the training data (Fig 3 bottom left, gray). The optimal regularization strength was selected using the validation dataset by (i) Bayesian Information Criterion (BIC) computed on the training set (Fig 3 top right, gray) and (ii) by residual sum of squares (RSS) (Fig 3 bottom left, green). This regularization parameter was determined to be $\lambda^* = 3$ for both the BIC and RSS metric (Fig 3, bottom left). The same result was obtained when using Pearson's correlation coefficient (S4 Fig). We studied the

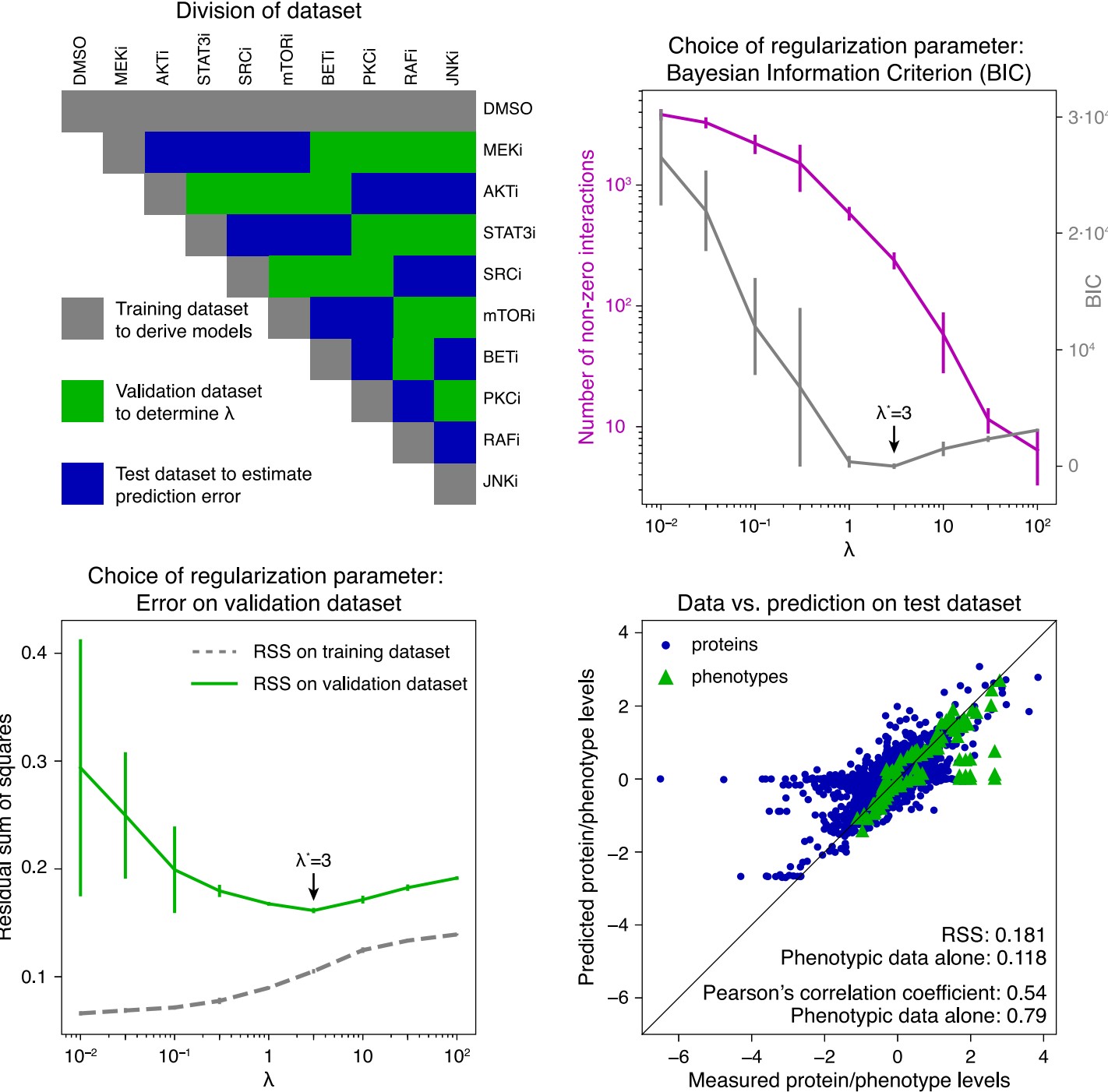

**Fig 3. Model selection and error estimation.** Top left: The full dataset was divided into subsets: (i) a training dataset (gray) that contained single drug control measurements (DMSO), single drugs in low (first row) and high dose (diagonal, two times low dose), (ii) a validation dataset (green) was used to estimate the optimal regularization parameter $\lambda^*$, and (iii) a test dataset (blue) was used to estimate model performance. Top right: calculated values for the Bayesian Information Criterion (BIC, gray) and number of non-zero interactions (magenta) as a function of the regularization parameter $\lambda$. Bottom left: The residual sum of squares on the validation dataset was used to identify the optimal regularization parameter $\lambda^*$. The best predictive model was obtained for $\lambda^* = 3$ according to lowest BIC and minimal error on the validation dataset. Error bars indicate the standard deviation from 10 independent runs. Bottom right: Agreement of measured and predicted protein and phospho-protein (dots) and phenotype levels (triangles) on the test dataset. The Pearson correlation coefficient on left-out data for the combined set of molecular and phenotype nodes is 0.54, and 0.79 for phenotypic nodes alone. The mean RSS for the combined phenotypic and molecular nodes is 0.181, and 0.118 for the phenotypic nodes alone.

accuracy of the final models on the independent test dataset (Fig 3 bottom right). The resulting generalization error for optimal $\lambda^*$ on left-out data had a mean of 0.181 across the 10 models and of 0.118 for phenotypic data alone, and the corresponding Pearson correlation coefficient between model prediction and the experimentally measured data was 0.54 (molecular and phenotypic) and 0.79 for the phenotypic data alone. The correlation was found to be higher in the later time points of molecular and phenotypic data. In particular, the Pearson correlation coefficient between predictions and measurements of the combined molecular and phenotypic data was 0.71 on the combined last three time points (24, 48, and 67 hours) and 0.74 for the last time point alone (S5 Fig). Hence, the inferred temporal model had the highest accuracy at the 67-hour time point. We conclude that the method for network modeling has a reasonably accurate predictive power, especially for predictions in the 24–67 hour range and that the models are useful for predicting the effects of new perturbations as hypotheses. Reasonable accuracy means that an affordable number of experiments would lead to a positively validated result (a 'hit') that can be advanced to pre-clinical investigation.

**Analysis of network models.** We used the optimal regularization parameter $\lambda^* = 3$ to infer 101 network models on all available data. To have reasonable diversity, the models were inferred without using prior information, i.e., known biological interactions. Several key model interactions agreed with interactions reported in the literature. The inferred effect of selected drugs on proteins (Fig 4) was in agreement with known drug–protein interaction patterns (e.g., MEKi inhibits the phorphorylation of ERK1/2 at T202/Y204 and RAFi inhibits MEK1/2 phorphorylation at S217/221) as well as unknown interactions (e.g., PKCi inhibits phorphorylation of CREB at S133). In addition, these model-derived drug–protein effects were in agreement with the observed drug effects from single-drug RPPA measurements (S2 Fig).

Saturation effects of drugs are taken into account by the model using the open parameter $\delta_i$. For values of $0 < \delta_i < 1$, we have an almost linear drug effect; values of $\delta_i > 1$ do not result in any significant change as drug concentrations increase. For the case of PKC and SRC inhibitors, we find a $\delta_i$ in the linear range, i.e., that a doubling of the low dose results in almost the double response, while RAF and JNK inhibitors have almost reached saturation, i.e., no significant difference between the drug effect in low and high dose of the drugs (S3 Fig).

**Predicted effect of drug perturbations and nomination of targets for drug testing.** Using the 101 inferred network models, we predicted the phenotypic response (cell growth

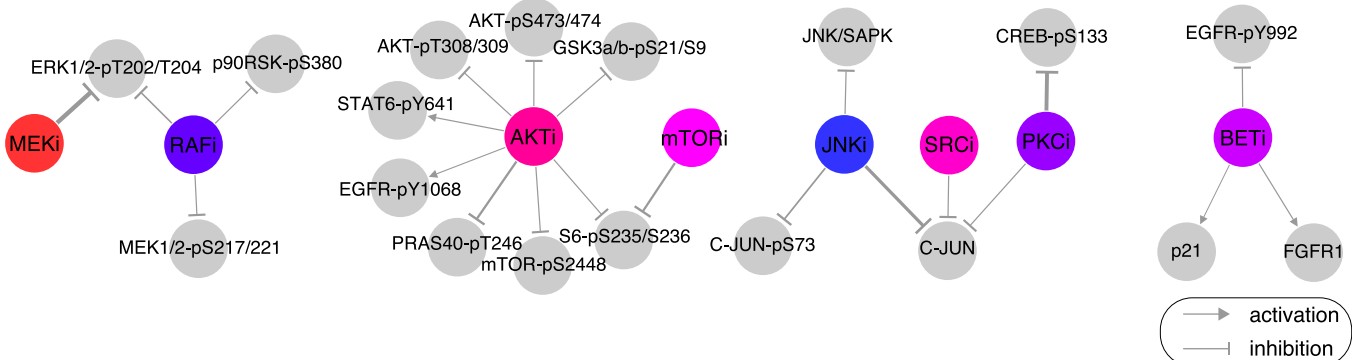

**Fig 4. Model-inferred effect of drugs on proteins and phospho-proteins.** The effect of drug treatment on (phospho-)protein levels is captured as edges between drugs (colored circles) and proteins (gray circles) in the model (represented by the drug–(phospo-)protein interaction strength $d_{il}$ from Eq 2). Some of the edges are well known (e.g., MEKi inhibits ERK1/2-pT202/T204) and some appear to be novel or indirect (e.g., inhibitory effect of PKCi on CREB-pS133). Based on the distribution of edge values over the 101 network models, only the strongest drug–protein edges are displayed for visualization purposes (85th percentile for positive/activating interactions and 15th percentile for negative/inhibiting interactions, by absolute value).

and apoptosis) as a function of network node inhibition (proteins and phospho-proteins). In particular, we systematically simulated a wide spectrum of inhibition strengths ($c^{pert}$) targeting every molecular node (Eq 7) and the phenotypic response according to Eq 8 at $t = 72$ h (see Materials and methods). The mean resulting dose response curves across all models when inhibiting each molecular node was used to estimate the half maximal effective concentrations ($EC_{50}$) for cell growth (S6 Fig) and apoptosis (S7 Fig). We further used these $EC_{50}$ concentrations to simulate the effect of pairwise combination perturbations of the 124 proteins and phospho-proteins on each phenotypic node independently across all models.

The predicted phenotypic responses were averaged across all models and the first 20 unique nodes were selected from the highest ranking combinations (Fig 5 left: cell growth reduction, right: apoptosis increase). In total, we predicted cell growth and apoptosis for $124 \cdot 123 = 15,252$ conditions for 101 networks, resulting in 1,540,452 simulated responses (S8 and S9 Figs).

The most effective predicted combination for decreasing cell number was found to be inhibition of phospho-tyrosine at position 992 on EGFR together with phospho-serine at positions 636/639 on IRS1 (Fig 5 left). Simultaneous inhibition of the protein IRS1 together with reduction of phospho-serine at positions 636/639 on IRS1 was found to the most effective combination for increasing apoptosis (Fig 5 right). Note that predicted perturbation of IRS1 alone had the nearly the same effect on apoptosis as in the combination.

EGFR is involved in growth signaling through the RAS/MEK pathway, and EGFR-pY992 is an activating auto-phosphorylation site at the receptor [22]. There are many available drugs that specifically inhibit EGFR and are expected to decrease EGFR-pY992, for example the drugs lapatinib [23], gefinitib, and erlotinib. Inhibitors of EGFR alone do not have a strong anti-proliferative effect in melanoma cell lines [24].

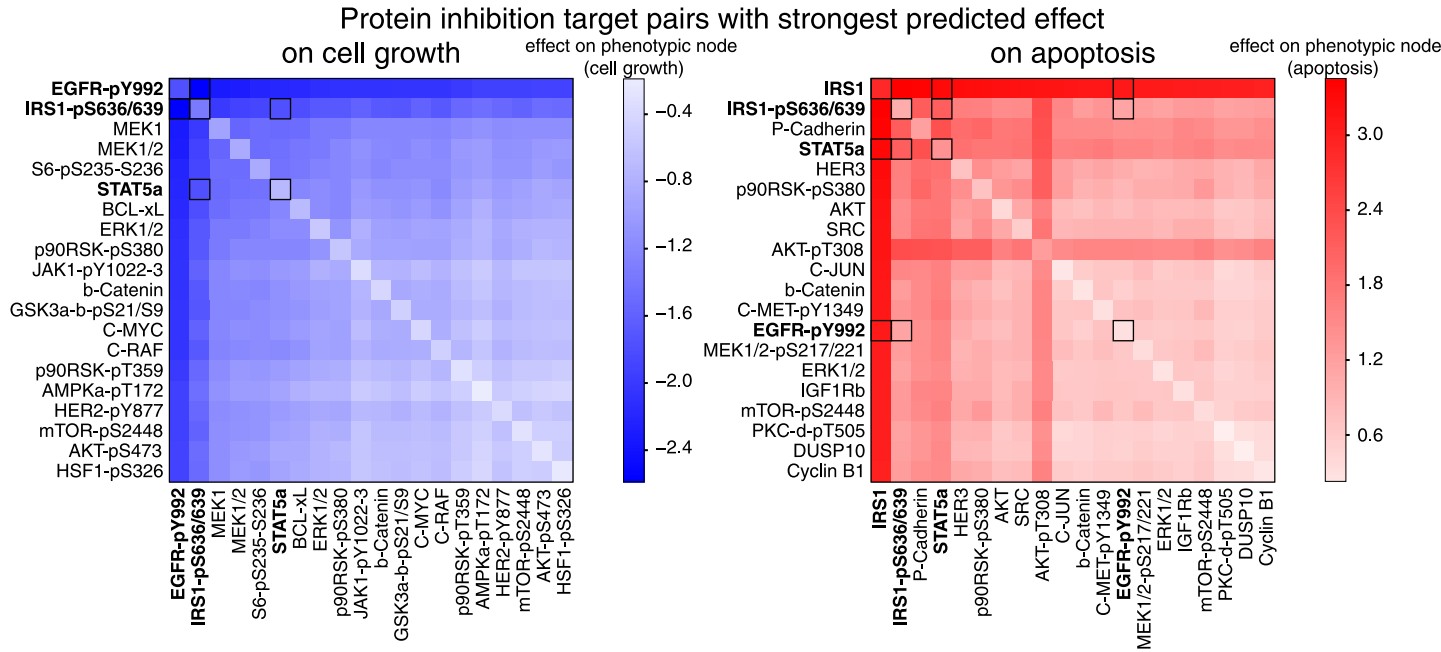

**Fig 5. Model predicts the effect of combination perturbations and suggests optimal inhibitor combinations.** The top 20 × 20 predictions of pairwise inhibition of molecular nodes (i.e., proteins and phospho-proteins) that decrease cell growth (bottom left, blue) and increase apoptosis (bottom right, red). Cell growth and apoptosis were computed for each target combination. These values were log₂-transformed, normalized to the unperturbed steady state, and the average value over 101 network model predictions is presented (diagonal represents predictions for inhibition of single targets). Combinations nominated for drug testing are highlighted by dark-rimmed squares. For complete heatmaps of all tested predictions, see S8 and S9 Figs.

IRS1 transmits signals from IGFR to PI3K/AKT. The protein can be inhibited by the compound NT157 [25], which has two mechanisms of action: 1) reduce IGF1 signaling through dissociation of IRS1 from the IGF1 receptor, and 2) reduce levels of IRS1 by phosphorylation at serine sites that lead to proteosomal degradation [25]. IRS1-pS636/639 is one such serine phosphorylation site [25]. The inhibitor NT157 is reported to first increase IRS1-pS636/639, which induces degradation of IRS1 (within hours) [25]. It has been reported that NT157 inhibits proliferation in the RAF inhibitor resistant melanoma cell line A375, both through reduced IGF1-signaling and reduced STAT3-signaling [26]. The inhibitor NT157 has also been shown to increase the tyrosine-phosphorylation 1172 at EGFR in the melanoma cell line A375 [27]. Using an EGFR inhibitor to abrogate this increase in EGFR-pY1172 after NT157 treatment may provide a mechanism for increased drug efficacy. In summary, simultaneous inhibition of IRS1 and EGFR is predicted to both reduce cell growth and increase apoptosis. In theory, this effect can be achieved by using the drugs NT157 (IRS1 inhibitor) and gefitinib (EGFR inhibitor).

## Experimental testing of model predicted drug effects

To test the predictive potential of our model, we selected a subset of the 124 molecular nodes for experimental testing. Nodes were chosen based on i) those that had the strongest predicted effect i.e., EGFR-pY992 and IRS1-pS636/639, ii) those that cover a wide range of predicted response patterns from substantial growth reduction to no effect on growth (Fig 6 bottom), and iii) those that were targetable by available drugs (Fig 6 top). Based on these criteria, we selected gefitinib and NT157 to target EGFR-pY992 and IRS1-pS636/639, respectively. The drug CAS285986, which inhibits STAT5b, was chosen as a proxy for STAT5a inhibition, and the p38-MAPK inhibitor SB203580 was selected to target p38-MAPK-pT180/T182. We applied seven different concentrations of each drug, alone and in selected pairwise combinations (Fig 6), and measured cell number in three-hour steps from time 0 until 96 h in three replicates (Materials and Methods).

The drug pair with the strongest predicted effect, NT157 and gefitinib, reduced cell growth substantially relative to no-drug control, and the combination is more effective than the effect of either drug alone (Fig 6 middle). The drug pair predicted to have the second strongest effect, NT157 and CAS285986, also reduced cell growth, but to a lesser extent than NT157 with gefitinib, and only for drug combinations that included the highest dose of NT157 (10 $\mu$M) (Fig 6 middle, 2nd row). NT157 and gefitinib were both predicted to reduce growth alone, and experimental tests resulted in a reduced cell number with the highest dose tested (10 $\mu$M) (Fig 6 middle, 2nd row). CAS285986 alone was predicted to cause a slight reduction in cell number, and no effect was observed at any dose of the drug (Fig 6 middle, 3rd row). SB203580 was predicted to increase growth, but no effect of the drug was seen at any concentration (Fig 6 middle, 3rd row). Although we predicted the combination of NT157 and SB203580 would have little-to-no effect on cell growth (with SB203580 antagonizing the effect of NT157), the experimental data did not support this hypothesis. However, a substantial reduction at the highest dose of NT157 (10 $\mu$M) was measured (Fig 6 middle, 1st row). The combination of gefitinib and CAS285986 failed quality control and results were inconclusive. In summary, with the exception of the NT157–SB203580 combination, the experimental results largely agreed with the model-based predictions.

## Comparison with literature data

We compared our predictions for cell growth with measured drug sensitivities from literature. From the rich perturbation dataset of [28] we selected those data that (1) referred to A2058

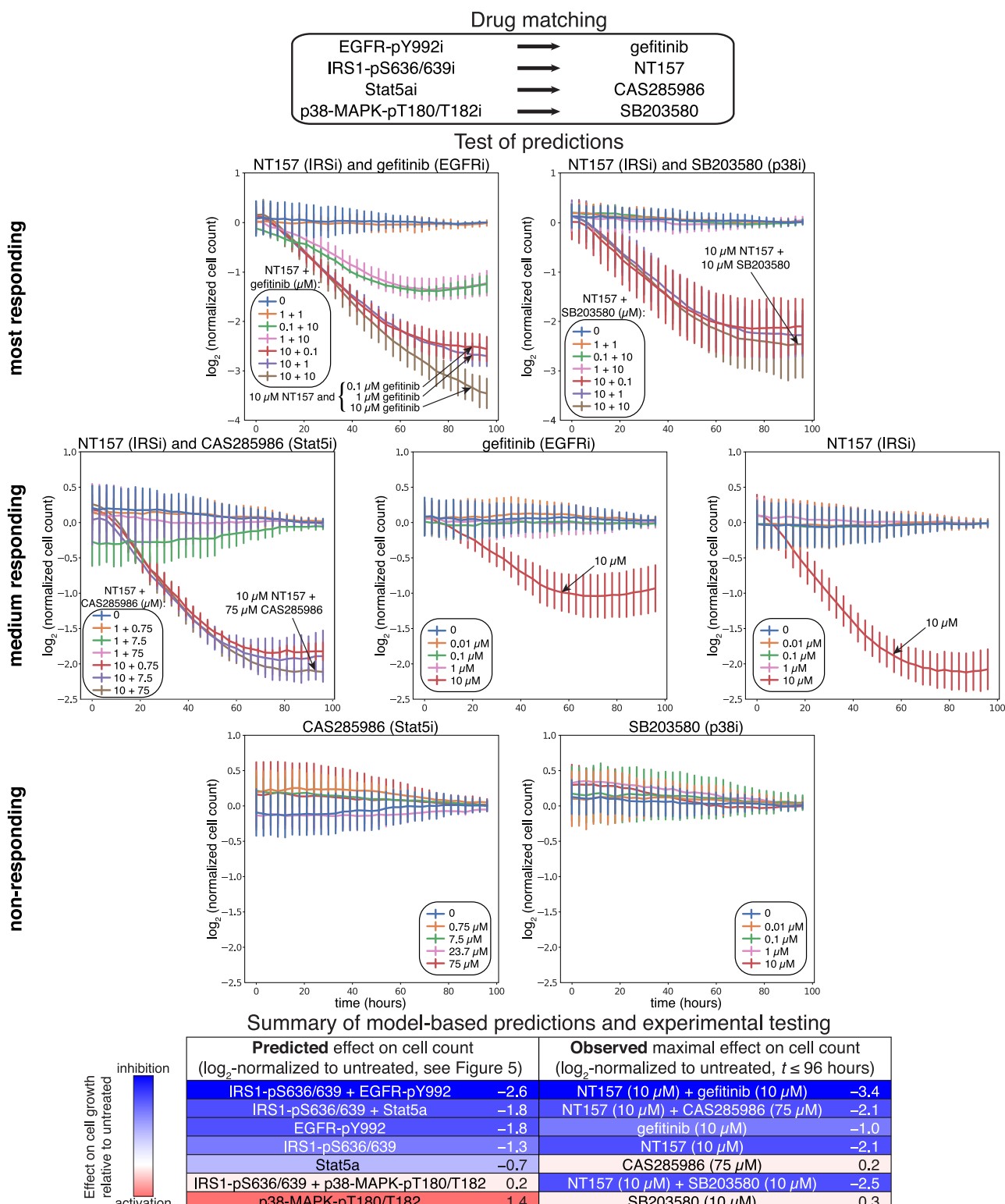

**Fig 6. Experimental testing of model predictions using single and pairwise drug combinations.** Several single and pairwise perturbations that were predicted to have a differing phenotypic effects were experimentally tested in the melanoma cell line A2058 (highlighted in Fig 5). Cells were perturbed with drugs that target nodes in the computational model (top). During perturbation, cells were subjected to live-cell imaging, the resulting images were segmented, and cell count was quantified, normalized relative to no-drug control and log2-transformed (see growth curves in middle panel). Agreement between model prediction (log2-transformed, normalized and averaged growth) and experiment is summarized (bottom panel).

and (2) applied drug perturbations that specified target proteins in our model (and measured input dataset). The selected data and the corresponding model predictions are shown in S2 Table. Many of our predictions of drug effects on cell growth are in agreement with previous measurements, e.g., ERK is highlighted as a potential target in both the experimental data and in our predictions. Further, we do not observe any gross false positive predictions in this comparison, i.e., where the model predicts perturbation of a protein target to be effective, and the experimental data show the opposite. However, we do observe several discrepancies between model prediction and experimental evidence in the literature, e.g., the inhibition of the receptor FGFR1 is predicted by our models to *increase* cell growth, while the measurements in [28] show that a drug that targets FGFR1 results in decreased cell growth. Comparison of model predictions and experimental drug sensitivities is shown in S2 Table and S10 Fig.

## Discussion

We present a method to infer network models from time-resolved molecular and phenotypic perturbation biology data. These data consist of multiple drug perturbations, applied as single drugs and drug combinations in a melanoma cell line. The computational method utilizes the full time-series data, and outputs network models that characterize the interactions between drug perturbations, (phospho-)proteins, and phenotypic changes. The inferred models allow for the prediction of phenotypic responses—cell growth and apoptosis—for unseen perturbations, and can therefore be used to generate large-scale drug discovery hypotheses. As an application, we use the method to predict effective drug target combinations in melanoma. In order to estimate the confidence of the proposed approach, we experimentally test the effect of a subset of the nominated drug combinations on cell growth. Given the complexity and resource requirements of experimental testing, we believe that this method provides a rational means to test many more perturbations than can be experimentally explored.

Inhibition of proteins EGFR and IRS1, which was predicted to be the most effective combination, resulted in a reduced number of cells only at high doses (10 $\mu$M) of the selected drugs. There are several possible reasons that a high dose was required to reduce cell growth even though the model predicted the drugs to be very effective. First, drug mode-of-action is likely different than inhibiting proteomic nodes in the computational models. For example, the drug gefitinib inhibits the entire tyrosine kinase domain rather than the single modeled phosphorylation EGFR-pY992. There are several other EGFR-related nodes in our network models. For instance, EGFR-pY845 is predicted to *increase* cell number when reduced (S6 Fig). This difference in predicted effect between inhibiting EGFR-pY845 and EGFR-pY992 might hint at a stronger growth reduction of more specific downstream targets when EGFR is phosphorylated at Y992. Second, antibody cross-reactivity may be introducing noise in our training data leading to inaccuracies in the final predictions. For example, the EGFR-pY992 antibody can also react with HER2, and therefore the values of EGFR-pY992 in the training data may be confounded by HER2 expression. The drug we tested, gefitinib, is not a dual EGFR–HER2 inhibitor. There are other drugs that simultaneously inhibit both EGFR and HER2 (e.g., erlotinib or peletinib) and are more efficient in melanoma cell lines [24]. Third, although we attempted to select drugs that would mimic model node perturbation, the actual molecular effect of each drug in our cell line may not alter the same nodes as in the model. For example, the model predicted that reduction in IRS1-pS636/639 would alter cell growth, but the drug that we used to target IRS1 (NT157) may not decrease this phosphorylation site. In fact, NT157 has been shown to *increase* serine phosphorylations that mark IRS1 for proteosomal degradation in the melanoma cell line A375 [25]. Even though NT157 decreases IRS1 protein, we have no knowledge of whether NT157 has any effect on

IRS1-pS636/639. For these reasons, the model-based predicted vulnerability—inhibition of EGFR-pY992 and IRS1/IRS1-pS636/639 in A2058 cells—might be more potent when alternative drugs are chosen or with other methods of perturbation. Of note, perturbations including the p38 and MAPK inhibitor SB203580 showed the strongest disagreement between prediction and experimental validation. In particular, SB203580 alone did not increase cell growth, and significantly reduced cell growth in combination with NT157. We do not have a data-driven hypothesis for this discrepancy, but speculate that it could be due to off-target effects.

Our modeling framework is data-driven, which means that network interactions are directly inferred from data without the use of prior knowledge on drug–protein, protein–protein, and protein–phenotype interactions. We believe that data-driven discovery of drug–protein interactions is an important novelty as many drugs have unspecific or even unknown molecular activity. Even kinase inhibitors—which are commonly considered specific and are therefore used to draw conclusions about the physiological roles of molecules—have been shown to have significant off-target specificity. For example, when panels of kinase inhibitors have been profiled against panels of kinases, multiple on- and off-target effects were found [29–32]. Our framework is useful regardless of the specificity of inhibitors/drugs since specificity is "learned" and all perturbation data can be utilized as information for model training.

As the predictions are inferred from context-specific data without any use of prior knowledge, it is tempting to fully assess model accuracy by comparing inferred parameters to interactions found in the literature. However, a full quantitative comparison between network models and literature is difficult for several reasons: (1) The relationship between model transfer function and experimental dose response is not straightforward to establish as their shape and magnitude are unique to each modality. (2) Our models are derived in a single experimental context (e.g., A2058 cells after perturbation in our hands), while literature contains an aggregate of knowledge across many different contexts from many different labs. (3) It is not rare that drugs that target the same molecule have opposing downstream effects, e.g., PI2K alpha inhibitors alpelisib and GNE-317 have positive and negative growth effects, respectively (see S2 Table). These opposing effects may be due to dose differences or incorrectly annotated specificity, both of which are difficult to tease apart. (4) The dynamic nature of drug response is often not well described in the literature and so full comparisons with our dynamic models are difficult or impossible. To facilitate a comparison, we generally access response at a steady state timepoint, which not correspond to literature-selected timepoints. (5) Although one can take many simulation strategies, our model simulation approach applied highly specific perturbations (dose and target) that would be impossible to achieve experimentally. (6) During model training, we took a simplified view of drug–target interactions, modeling the effect of each drug as a reduction in the activity of a single corresponding target. The relationship between drugs and targets is more complex and is often is many-to-many (see S2 Table). Even with these caveats, when well-established molecular interactions overlap with strong predictions, we feel confident that the model is capturing true biological relationships.

Similar to other data-driven modeling methods, our framework is heavily dependent on both the quality and quantity of data. We measured the response to 54 drug combinations at 8 time points, resulting in 432 data points for each node. Even with these rich data, we observed large differences in individual models in the prediction of new drug targets (S11 Fig). These results suggest that the data may not be sufficiently informative to generate unique models. To avoid mis-interpreting individual predictions, we believe it is important to study a representative set of network models, i.e., not only the single best solution. In this study, we generally aggregated the responses from 101 models. Another challenge for large models is determination of unique parameters. In principle, uncertainty of parameter determination can be

quantified using, e.g., profile likelihood assessment [33, 34], but this is computationally prohibitive for the relatively large models derived here.

The optimization problem dealt with in this study is of exponential complexity, with an exponent of $N^2$, where $N$ is the number of nodes. This severely limits the number of nodes that can be included in the model. Using the 124 (phospho-)protein nodes, each model requires inference of approximately 16,000 parameters, which required the use of high-performance compute facilities. In addition, the choice of nodes that are measured experimentally and used for model building is key to overall predictability and interpretability, and interactions between nodes should not necessarily be interpreted as direct biochemical interactions, but logical indirect interactions. In theory, more comprehensive and specific experimental measurements can be used to generate models that contain more direct interactions and predict with higher accuracy. Future studies will aim to generate such data using methods like CRISPR-Cas9 gene knockouts for perturbation and mass spectrometry for molecular characterization.

## Materials and methods

### Genetic features of melanoma cell line A2058

The selection of cell lines that are good models for observed alteration patterns in human tumor tissue is a non-trivial task [35]. In this study, we use the melanoma cell line A2058 for the derivation of optimal drug combinations based on patient-representative genomic profiles [36]. A2058 carries the BRAFV600E mutation that resembles the patient segment which is treatable with the RAF inhibitor vemurafenib and MEK inhibitors such as trametinib. However, A2058 is a RAF/MEK inhibitor resistant cell line with concurrent activation of ERK, PI3K/AKT and cell cycle pathways and TP53 mutation. Specifically, it carries BRAFV600E, MAP2K1P124S and TP53V274F mutations and is altered in tumor suppressor genes like CDKN2A, RB and PTEN [37]. As in most melanoma samples, the cell line A2058 carries a large number of additional genomic alterations other than those listed here [36, 38–40]. For A2058 cells, a synergistic effect of drug combinations has been shown using the combination of BRAF and PI3K inhibition (drugs: PLX4720 and GDC0941, respectively) [38].

### Model equations

For the molecular response model, molecular nodes represent concentrations of proteins and phospho-proteins. We start with Eq 1. Applied perturbations only affect molecular nodes. All interactions between molecular nodes are directional. Self-interaction terms are excluded from the model formulation. The resulting model of the temporal dynamics of the $i$-th **molecular node** $i = 1, \ldots, N_{\mathrm{molec}}$ in experimental condition $\mu$ is given by,

$$\frac{\mathrm{d}x_i^\mu(t)}{\mathrm{d}t} = \varepsilon_i \tanh\left(\sum_{j=1, j\neq i}^{N_{\mathrm{molec}}} w_{ij} x_j^\mu(t) + \sum_{l=1}^{N_{\mathrm{drug}}} d_{il} u_l^\mu(t)\right) - \alpha_i x_i^\mu(t). \tag{2}$$

In particular, $x_i^\mu(t)$ denotes the temporal log$_2$-control-normalized protein level (see defining equation in Data normalization below), which is subjected to drug perturbations as defined by the superposition of drug nodes $u_l^\mu(t)$, $l = 1, \ldots, N_{\mathrm{drug}}$ (see below). In order to model saturation effects and cap the time derivatives, we use the hyperbolic tangent as the sigmoidal transfer function. By doing so, the interaction-driven differential change of variable $i$ is bounded to values in $[-\varepsilon_i, \varepsilon_i]$. The interaction parameters $w_{ij}$ characterize the effect of protein level $j$ on $i$ and $\alpha_i$ describes how quickly the $i$-th protein level returns to the zero-level steady-state in the absence of any interaction or drug. The specific effect of the drug $l$ on molecular node $i$ is

quantified by $d_{il}$. In contrast to our previous perturbation modeling approaches [4, 21] in which the interaction between drug node and target were defined based on prior knowledge, in this work we determine the drug–molecular node interactions $d_{il}$ as a result of model inference. In addition to $d_{il}$, the parameters $\alpha_i$, $\varepsilon_i$ and $w_{ij}$ are also data-derived.

Using Eq 1, we define the $l$-th **drug node**, $l = 1, \ldots, N_{\text{drug}}$, by,

$$\frac{\mathrm{d}u_l^\mu(t)}{\mathrm{d}t} = 10 \cdot \tanh(\delta_l c_l^\mu) - 10 \cdot u_l^\mu(t), \tag{3}$$

where we enforce a fast-acting drug effect by choosing $\varepsilon = \alpha = 10$ in Eq 1. Here, $c_l^\mu$ describes the drug concentration relative to the single-drug concentration (i.e., $c_l^\mu = 0$ when the drug not present, $c_l^\mu = 1$ when drug is present at low dose concentration and $c_l^\mu = 2$ when the drug is present at high dose (i.e., $2 \times$ low dose) in experimental condition $\mu$. Eq 3 has the analytical solution,

$$u_i^\mu(t) = \tanh(\delta_l c_l^\mu)(1 - e^{-10t})$$

and $\delta_l$ is the effective impact, which is inferred in the parameter inference process.

We also use Eq 2 to define the two **phenotypic nodes** cell growth (cg) and apoptosis (ap). These nodes are modeled in the same way as molecular nodes except that interactions are only unidirectional (i.e., from molecular to phenotypic node). These dynamics are described as,

$$\frac{\mathrm{d}x_{\text{cg}}^\mu(t)}{\mathrm{d}t} = \varepsilon_{\text{gc}} \tanh\left(\sum_{j=1}^{N_{\text{molec}}} w_{\text{cg},j} x_j^\mu(t)\right) - \alpha_{\text{gc}} x_{\text{gc}}^\mu(t),$$

$$\frac{\mathrm{d}x_{\text{ap}}^\mu(t)}{\mathrm{d}t} = \varepsilon_{\text{ap}} \tanh\left(\sum_{j=1}^{N_{\text{molec}}} w_{\text{ap},j} x_j^\mu(t)\right) - \alpha_{\text{ap}} x_{\text{ap}}^\mu(t), \tag{4}$$

where $x_{\text{cg/ap}}^\mu(t)$ denotes the temporal $\log_2$-control-normalized apoptotic readout and cell number, respectively (see defining equation in Data normalization below).

## Definition of loss function and parameter estimation

The residual sum of squares for a given parameter set $\Theta$, RSS($\Theta$), is used as measure of agreement between model simulations and observations,

$$\mathrm{RSS}(\Theta) := \sum_\mu \sum_{i=1}^{N_{\text{meas}}} \sum_k \left(x_i^\mu(t_k; \Theta) - x_{i,k}^\mu\right)^2, \tag{5}$$

where all free parameters are grouped into $\Theta := \left((\varepsilon_i)_{i=1}^{N_{\text{meas}}}, (\alpha_i)_{i=1}^{N_{\text{meas}}}, (w_{ij})_{i,j=1}^{N_{\text{meas}}}, (d_{il})_{i=1,l=1}^{N_{\text{meas}}, N_{\text{drug}}} \times (\delta_i)_{i=1}^{N_{\text{drug}}}\right)$ and $N_{\text{meas}} = N_{\text{molec}} + N_{\text{phen}} = N_{\text{molec}} + 2$ is the number of measured observables, i.e., the number of proteins and phospho-proteins (molecular nodes), and the two phenotypes. Moreover, $x_i^\mu(\Theta; t_k)$ represents the simulated response given the parameters $\Theta$, $\mu$ is the experimental condition and $t_k$ is the $k$-th time point. The corresponding observed data is denoted by $x_{i,k}^\mu$. In addition to the minimization in Eq (5), regularization was applied to reduce the number of edges of non-zero edges. For this purpose, an $\ell^1$-norm on the interaction parameters $w_{ij}$ was

added to the problem of minimizing the residual sum of squares as function of $\Theta$:

$$\lambda \sum_{i=1}^{N_{\text{meas}}} \sum_{j=1, j\neq i}^{N_{\text{molec}}} |w_{ij}|, \tag{6}$$

which defines the to-be minimized loss function for our parameter estimation. Here $\lambda > 0$ sets the regularization strength and was chosen based on data holdout and Bayesian Information Criterion considerations.

The loss function was minimized using Adam, an implementation of a stochastic algorithm for first-order gradient-based optimization [41]. We used the version of Adam included in TensorFlow (version 1.2.1, Google Brain team, Mountain View, CA, U.S.A.), and the entire simulation and optimization procedure was performed in a Python environment. For all minimization runs, we used a fixed learning rate of 0.01 and evaluated the results when 10,000 iterations or 24 hour run time were reached, whichever occurred first. Failed optimization runs with all $w_{ij} = 0$ were discarded.

The Adam minimizer does not offer soft thresholding, so no exact zeros of parameters are obtained when the loss function (i.e., the sum of Eqs (5) and (6)) is minimized. We adopt an alternative strategy to introduce parameter sparsity by cropping the interaction parameter $w_{ij}$ in each update step $n \mapsto n + 1$, for interactions from protein/phenotypic nodes: $w_{ij}^{n+1} = 0$ if $w_{ij}^n < 0.01$, and for interactions from drugs nodes: $d_{il}^{n+1} = 0$ if $d_{il}^n < 0.05$. For our specific set-up, this strategy to enforce sparsity in the inferred parameters produced stable parameter estimates with a decrease in computational run time relative to full representations.

Recently, other perturbation biology modeling strategies based on ours and others [4, 21] have explored approaches for better scaling data-driven models using TensorFlow [42].

## Model selection

We adopted two strategies to select the most informative model without overfitting to the given data using (1) the Bayesian information criterion and (2) by division of the input data into training, validation, and test datasets.

**Bayesian information criterion.** Assuming independent and identically distributed model errors drawn from a normal distribution, the maximum loglikelihood $\ln(l(\hat{\Theta}))$ can be expressed in terms of the minimum $\text{RSS}(\hat{\Theta})$, where $\hat{\Theta}$ is the parameter set that maximizes the likelihood function. We use the following simplified definition of the Bayesian information criterion, BIC,

$$\text{BIC}(k) = -2\ln(l(\hat{\Theta})) + k\ln(n) \cong n\ln(\text{RSS}(\hat{\Theta})/n) + k\ln(n)$$

where $k$ is the number of non-zero model parameters and $n$ is the number of data points used in the parameter estimation. The number of non-zero parameters is a function of the regularization applied via $\lambda$, i.e., $k = k(\lambda)$. We select the optimal regularization parameter $\lambda$ as the minimum of the BIC,

$$\lambda^* = \arg\min_{\lambda} \text{BIC}(\lambda) \equiv \arg\min_{\lambda} \{n\ln(\text{RSS}(\hat{\Theta})) + k(\lambda)\ln(n)\},$$

where we omitted contributions from constants that are dependent on the number of data points $n$. For the current data set, the optimal regularization parameter is found as $\lambda^* = 3$ for the Bayesian information criterion (Fig 3 top right). The Akaike information criterion did not have a clear minimum.

**Dataset division.** In addition to selecting λ using the Bayesian Information Criterion, the optimal regularization parameter $\lambda^*$ is evaluated by splitting the data into a training dataset to estimate the model parameters, a validation dataset to determine an optimal λ, and a test dataset to evaluate model performance. The training set is chosen to contain all single drug conditions, and the test sets are chosen as a split between all combinations of drugs (Fig 3 top left). As with the Bayesian Information Criterion, the optimal regularization parameter is found to be $\lambda^* = 3$ (Fig 3 bottom left).

## Systematic *in silico* perturbation of molecular nodes

In order to identify new potential drug target combinations, we combine simulations using models built with all the data with an *in silico* inhibitor of each of molecular node, $x_i^{\text{pert}}(t)$. The *in silico* inhibitor of molecular node $i$ is added to the model equations as a step concentration $c_i^{\text{pert}} \geq 0$ for $t \geq 0$ and then we simulate the protein and phospho-protein dynamics to the last measured time point $t = 72$ h. The resulting dynamics for the perturbed/inhibited molecular node $i = 1, \ldots N_{\text{molec}}$ are found from,

$$
\begin{aligned}
\frac{dx_i^{\text{pert}}(t)}{dt} &= \varepsilon_i \tanh\left(\sum_{j=1, j\neq i}^{N_{\text{molec}}} w_{ij} x_j^{\text{pert}}(t) - c_i^{\text{pert}}\right) - \alpha_i x_i^{\text{pert}}(t), \\
\frac{dx_{i'}^{\text{pert}}(t)}{dt} &= \varepsilon_{i'} \tanh\left(\sum_{j=1, j\neq i'}^{N_{\text{molec}}} w_{i'j} x_j^{\text{pert}}(t)\right) - \alpha_{i'} x_{i'}^{\text{pert}}(t)
\end{aligned}
\tag{7}
$$

with $i' = 1, \ldots, i - 1, i + 1, \ldots, N_{\text{molec}}$. The response of the phenotypic nodes, $x_{\text{cg}}^{\text{pert}}(t)$ and $x_{\text{ap}}^{\text{pert}}(t)$, are determined as,

$$
\begin{aligned}
\frac{dx_{\text{cg}}^{\text{pert}}(t)}{dt} &= \varepsilon_{\text{gc}} \tanh\left(\sum_{j=1}^{N_{\text{molec}}} w_{\text{cg},j} x_j^{\text{pert}}(t)\right) - \alpha_{\text{gc}} x_{\text{gc}}^{\text{pert}}(t), \\
\frac{dx_{\text{ap}}^{\text{pert}}(t)}{dt} &= \varepsilon_{\text{ap}} \tanh\left(\sum_{j=1}^{N_{\text{molec}}} w_{\text{ap},j} x_j^{\text{pert}}(t)\right) - \alpha_{\text{ap}} x_{\text{ap}}^{\text{pert}}(t).
\end{aligned}
\tag{8}
$$

**Selection of inhibition strength for *in silico* combination perturbations.** We assume the solution of Eqs 7 and 8 at $t = 72$ h to have the following shape when molecular node $i$ is perturbed with a single *in silico* inhibitor applied with concentration $c_i^{\text{pert}} \geq 0$,

$$
x_{\text{cg/ap}}^{\text{pert}}(c_i^{\text{pert}}) = \frac{E_2 - E_1}{1 + \left(\frac{c_i^{\text{pert}}}{c}\right)^n} + E_1,
$$

where $x_{\text{cg/ap}}^{\text{pert}}(\infty) = E_1$ and $x_{\text{cg/ap}}^{\text{pert}}(0) = E_2$ represent the maximum and minimum change in each phenotype, respectively. $c_i^{\text{pert}}$ is the concentration of the *in silico* inhibitor specifically targeting molecular node $i$. $n$ and $c$ are open parameters. $n$ is the Hill coefficient and $c$ is the *in silico*-derived 50%-effect concentration, $EC_{50}$. In our framework, we focus on the effect on cell growth and apoptosis by single molecular node inhibition. As the data are all $\log_2$-normalized

to control data, $x_{\text{cg/ap}}^{\text{pert}}(0) = E_2 = 0$ and so the equation above simplifies to,

$$x_{\text{cg/ap}}^{\text{pert}}(c_i^{\text{pert}}) = E_1 \left( 1 - \frac{1}{1 + \left( \dfrac{c_i^{\text{pert}}}{c} \right)^n} \right) \tag{9}$$

with $x_{\text{cg/ap}}^{\text{pert}}(0) = 0$ and $x_{\text{cg/ap}}^{\text{pert}}(\infty) = E_1$ (see S6 and S7 Figs for the mean individual dose response of apoptosis and cell growth as a function of inhibiting single molecular nodes). These *in silico*-derived $EC_{50}$ are subsequently used for molecular node inhibition simulations. These simulations are systematically carried out such that all single and pairwise molecular nodes are perturbed, and their effect on cell growth and apoptosis is calculated (Fig 5).

The scale of concentration in the simulations, and hence the estimated *in silico* $EC_{50}$ concentration (parameter $c$ in Eq (9), S6 Fig), is expressed in arbitrary units, but can nevertheless be related to the experimental $EC_{50}$ concentration. This can be done by equalising the estimated *in silico* and experimentally determined $EC_{50}$ values. This assumes however that the inhibitor used in the experiments is of perfect specificity.

**Numerical solution and simulation of model equations and systematic perturbations.** The above specified ordinary differential equations were numerically solved with a second-order Runge–Kutta method with two stages and a fixed step size in a Python environment. As a quality check, these results were compared with those using the ode15s solver for stiff differential equations in MATLAB R2018a (Mathworks, Natick, MA, U.S.A.) and were in agreement.

## Experimental and data normalization protocols

**Molecular measurements in response to perturbation.** The melanoma cell line A2058 was seeded in 6-well plates at 50,000 cells per well. Each biological replicate was run on separate occasions. The physical difficulties and logistical complexity of the experiments required that each time point be run in batches with the early time points being harvested immediately and the late time points being setup immediately following time point harvesting. Perturbed cells were lysed in CLB1 buffer (Bayer Technology Services, Leverkusen, Germany; now NMI TT Pharmaservices, Reutlingen, Germany). For Dataset 1, we performed a 4-fold dilution series of the samples using the Biomek FXP Laboratory Automation Workstation (Beckman Coulter Inc., Brea, CA, U.S.A.) automatic pipetting system in one technical replicate [19]. For Dataset 2, [a single lysate sample was printed twice (technical replicate)] each lysate sample was printed at the respective protein concentration in two technical replicates. Diluted samples were printed onto tantalum pentoxide-coated glass chips and then blocked with an aerosol BSA solution. The protein array chips are then washed in double-distilled $H_2O$ and dried before measurement. For the immunoassay, we incubated the chips with primary antibodies for 24 hours followed by 2.5 hours incubation with Alexa Fluor-647 conjugated secondary antibody detection reagents. Antibodies are diluted in CAB1 buffer (Bayer Technology Services, Leverkusen, Germany). The immuno-stained chips were imaged using the ZeptoREADER (Zeptosens/Bayer, Witterswil, Switzerland). The ZeptoView 3.1 software (Zeptosens/Bayer, Witterswil, Switzerland) was used to output the reference net spot fluorescence intensity, RNFI. Included standard global and local normalization of sample signal to the reference BSA grid is used.

**Sample lysis and RPPA chip layout and analysis.** All experimental perturbations and sample preparations were carried out in the Sander lab. Cell lysate was aliquoted into multiple

samples, and RPPA measurement using the zeptosens platform was carried out in both the Sander lab (MSKCC, Dataset 1) and the Pawlak lab (NMI TT Pharmaservices, Dataset 2).

**Differences between each dataset.** Dataset 1 consists of 71 antibody measurements of three biological replicates at eight time points (10 min, 27 min, 72 min, 3 h, 9 h, 24 h, 48 h, 67 h). For Dataset 1, each RFI data point, $\hat{x}_{i,k}^{\mu}$, was derived from a four-spot protein dilution series of 0.2, 0.15, 0.1, and 0.05 $\frac{\text{mg}}{\text{ml}}$. Dataset 2 consists of 86 antibody measurements of three biological replicates at (a) eight time points for biological replicate 1 and (b) two time points (48 h, 67 h) for biological replicate 2 and 3. For Dataset 2, protein lysate was diluted to 3 $\mu$g/$\mu$l, when concentration was higher than 3.4 $\mu$g/$\mu$l, and each RFI data point was derived from the mean of two measured spots. Median-centered protein factors, determined for each sample by NMI TT Pharmaservices, were used to normalize protein content.

**Similarities between each dataset.** All data were acquired from the same RPPA platform (Zeptosens/Bayer, Witterswil, Switzerland). 33 antibodies were measured in both datasets as controls, which resulted in 71 + 86 − 33 = 124 unique antibodies. After outlier detection and loading normalization (see above), the median of the three biological replicates from Dataset 1 and the median of the data (one biological replicate in six time points and three biological replicates in two time points) from Dataset 2 was combined and used in the downstream analysis and modeling steps.

**Fluorescence signal evaluation.** Each spot measured on the Zeptosens array corresponds to the relative concentration of a specific protein or phospho-protein in a single condition. This value is detected by measuring the secondary antibody fluorescence that is bound to the (phospho)protein-specific antibody. The relationship between this signal and the absolute concentration of the (phospho-)protein is assumed to be linear in the observed range, i.e.,

$$c_i \sim \hat{x}_i$$

given constant exposure times across all the experimental conditions and time points in each antibody measurement. In Dataset 1, for each (phospho-)protein $i$ one measures four antibody fluorescence signals $\hat{x}_i^{(1)}$, $\hat{x}_i^{(2)}$, $\hat{x}_i^{(3)}$ and $\hat{x}_i^{(4)}$, corresponding to the four serial dilutions of cell lysate $c_i^{(1)} = c_i$, $c_i^{(2)} = \frac{3}{4} c_i$, $c_i^{(3)} = \frac{1}{2} c_i$ and $c_i^{(4)} = \frac{1}{4} c_i$. These values are referred to as Referenced Net spot Fluorescence Intensity (RNFI). This allows to quantify the degree of "non-linearity" of the signal by computing the discrepancy between linear fit and actual signal. Assuming a linear dependency, then $\hat{x}_i(c) = \beta_0 + \beta_1 c$, the fitting parameters are found in a least-squares estimation,

$$(\beta_0, \beta_1)^{\mathrm{T}} = \underset{\beta_0, \beta_1}{\arg\min} \sum_{m=1}^{4} \left| \hat{x}_i^{(m)} - \left( \beta_0 + \beta_1 \frac{m}{4} c_i \right) \right|^2 = (\mathbf{c}^{\mathrm{T}}\mathbf{c})^{-1}\mathbf{c}^{\mathrm{T}}\mathbf{y} = \begin{pmatrix} 1 & 0.5 & 0 & -0.5 \\ -\dfrac{6}{5c_i} & -\dfrac{2}{5c_i} & \dfrac{2}{5c_i} & \dfrac{6}{5c_i} \end{pmatrix} \mathbf{y},$$

where,

$$\mathbf{c} = \begin{pmatrix} 1 & 0.25c_i \\ \vdots & \vdots \\ 1 & c_i \end{pmatrix}, \qquad \mathbf{y} = \begin{pmatrix} \hat{x}_i^{(1)} \\ \vdots \\ \hat{x}_i^{(4)} \end{pmatrix}.$$

For ideal profiles, i.e., without intercept or $\hat{x}_i(0) = \beta_0 \approx 0$, we have $\hat{x}_i(c_i) \approx \beta_1 c_i$ (and consequently, $\frac{\hat{x}_i(c_i)}{c_i} \approx \beta_1$). A single value, the Referenced sample Fluorescence Intensity (RFI), corresponding to the linear fit of the 4-spot array corresponding to the mean concentration value,

$c = 0.625\bar{c}$, is chosen,

$$\hat{x}_i := \hat{x}_i(0.625\bar{c}) = \beta_0 + \beta_1 \cdot 0.625\bar{c}.$$

Zeptosens calls the RFI using the reference concentration of the total protein content per spot ($\bar{c} = 0.2 \frac{mg}{ml}$). This concentration was chosen to be roughly the protein content of a eukaryote cell. Ideally, the concentrations for each sample are $c_i \approx \bar{c}$, but in practice $c_i$ varies due to experimental uncertainties. We could estimate *a posteriori* the actual protein loading quantification with the reference loading chip and subsequently adjust $\hat{x}_i$ for each sample by setting $\bar{c} = \text{median}_i\{c_i\}$. For Dataset 2, we adopt this strategy and use on-chip measured median-centered protein factors determined from protein stain assays. However, for Dataset 1 we had no measurements of the protein loading and therefore applied double-median normalization (see [39]) for protein loading normalization.

**Spotting error detection.** For Dataset 1, due to stochastic sampling, low concentrations or saturated sample, as well as occasional robotic errors, some lysate spots in the serial dilutions do not follow the expected linear and monotonically decreasing dilution profile. We identify outliers that deviate strongly from the linear interpolation of the RNFI values using the Cook's distance [43]. We then use a Cook's distance of 0.8 as cutoff. For Dataset 2, the spotting quality appeared to be more consistent and no spotting error removal was required.

**Protein loading normalization.** Occasionally, samples have total protein concentrations that deviate from the optimal loading concentration. To normalize for this uneven loading in the we perform double-median normalization (sometimes referred to as "loading control") as described in [44]. For Dataset 1, the $\log_2$-transformed and normalized value for the $i$-th antibody in time-point $k$ and under experimental condition $\mu$ is then calculated as,

$$x_{i,k}^{\mu} := \frac{\log_2 \hat{x}_{i,k}^{\mu} - \text{med}_i}{\text{mad}_{k,\mu}}.$$

Here, $\hat{x}_{i,k}^{\mu}$ denotes the linear raw data of antibody $i$ measured in time point $k$ under experimental condition $\mu$, $\text{med}_i := \text{median}_{k,\mu}(\log_2(\hat{x}_{i,k}^{\mu}))$ the median of the $\log_2$-transformed values in each antibody across time points and experimental conditions and $\text{mad}_{k,\mu} := \text{median}_i(\log_2(\hat{x}_{i,k}^{\mu}) - \text{med}_i)$ its median absolute deviation. In Dataset 2, in contrast to Dataset 1, we have an estimate on the relative deviation from the target protein loading concentration in each sample of 0.3 $\mu$g/$\mu$l. The observed protein loading was lower and found to have a median concentration of 0.22 $\mu$g/$\mu$l). Based on these measurements, we normalize each lysate sample by the corresponding on-chip measured median-centered protein factors.

**Data normalization.** Before applying any further modeling and inference, we subtract the $\log_2$-transformed and double-median normalized RPPA data in the DMSO control condition (without perturbation) from all previously $\log_2$-transformed and double-median normalized RPPA data, $x_{i,k}^{\mu}$:

$$x_{i,k}^{\mu} \mapsto x_{i,k}^{\mu} - x_{i,k}^{\text{DMSO}}.$$

## Phenotypic measurements in response to perturbation

At the start of the experiment, 1,000 A2058 cells were seeded in three biological replicates in 96-well plates. After 24 hours, drugs were added to each well and cells were subjected to real-time imaging using the IncuCyte (Essen BioScience, Ann Arbor, MI, U.S.A.) live-cell imaging system. Images were taken every 3 hours in 3 channels (phase, GFP, RFP) at 4 regions on the plate (12 images per well at each time point). The RFP channel was used to detect nuclear-

localized mCherry (constitutively expressed). To detect apoptosis, a Caspase-3/7 green reagent (Essen BioScience) was added to each well. Image analysis using the Incucyte software was used to segment and count all cells (RFP channel) and cells going through apoptosis (GFP channel) at each time point. These counts were $\log_2$-transformed and normalized relative to DMSO control condition.

## Cell number measurements of optimal drug combinations

**Cell line.**  Cell line A2058 with H2B-mCherry was thawed, tested for mycoplasma, and passaged twice prior to testing. The cell line was passaged fewer than 15 times and screened with the compounds within a month of thawing. The cell line was passaged and cultured on DMEM media with 10% FBS and 1% Penicillin/Streptomycin.

**Screening protocol.**  Cells were harvested, counted, and deposited into 384 well plates at a total volume of 50 $\mu$l per well with a $t_0$ seeding density of 750 cells/well using a Thermo Multi-drop Combi dispenser (Thermo Fisher Scientific, Waltham, MA, U.S.A.). Cells were immediately placed into the Incucyte microscope culture and allowed to attach and proliferate for 24 hours. Four compounds (NT157, CAS285986, gefitinib, SB203580) were dissolved into 10 mM solutions two hours prior to dosing. The compounds were dispensed into the previously seeded 384 well plates using a HP D300e Digital Dispenser 24H after cell seeding in a dose matrix testing each concentration point of each drug in combination with each concentration point of the other drugs, with three of the drugs (NT157, gefitinib, and SB203580) titrated from 0.01 $\mu$M to 10 $\mu$M and CAS285986 titrated from 0.075 $\mu$M to 75 $\mu$M. The plates were then placed back into the Incucyte and cell measurements were recorded every 3 hours for 96 hours.

## Supporting information

**S1 Table. The drugs applied in this study together with each drugs' literature-described downstream target(s), the downstream target(s) measured in this study, and literature references.**
(XLSX)

**S2 Table. Comparison of literature-derived effect of single drugs on cell growth to model predictions.** A comparison of drug sensitivity data from the A2058 melanoma cell line [28] is based on known target proteins of each drug from [28]. Positive values represent an increase in growth when the drug is given (highlighted in gray). The corresponding model-predicted drug pair values are from the diagonal of S8 Fig.
(XLSX)

**S1 Fig. The simulated effect of each drug and drug pair on cell growth and apoptosis.** After 72 hours of treatment drug treatment, cell count and apoptosis was measured using live-cell imaging (Incucyte). Color intensity reflects cell count (left) and apoptosis (right) relative to untreated cells ($\log_2$-normalized). The cellular response to single drugs in both high (2 × low dose) and low dose as well as to all pair drug combinations (in low doses) were measured. The low dose effect of the drugs are depicted in the first row/column, and the high dose in the diagonal.
(EPS)

**S2 Fig. Drug responses of proteins and phospho-proteins.** For each drug, the six (phospho-) proteins depicted are those that exhibited the largest magnitude of response to single drug perturbations. The data is ranked by the absolute median response over time.
(EPS)

**S3 Fig. Temporal patterns of drug node dynamics.** The means and standard deviations of the simulated drug nodes for the high dose (solid line) and low dose (dashed line) of several *in silico* inhibitors across the 101 created network models.
(EPS)

**S4 Fig. Model selection and error estimation.** Mean and standard deviation of computed correlations for the validation dataset as a function of the regularization parameter λ. In agreement with the previous analysis, the best predictive model is obtained for $λ^* = 3$. Error bars indicate the standard deviation from 10 independent runs. Related to Fig 3.
(EPS)

**S5 Fig. The correlation between model simulation and experimental data.** Comparison between prediction and experiment for the last three measured time points, 24, 48, and 67 hours, (left) and for the last measured time point alone, 67 hours (right). This result, compared with Fig 3, suggests that the model predictions are less reliable in earlier time points, potentially due to the transient nature of the drug response and / or experimental noise at earlier time points in the data.
(EPS)

**S6 Fig. The effect on predicted cell growth due to single node inhibition.** All individual network model were simulated with different levels of input strength of an *in silico* inhibitor for each target present in the model. From these simulations, the mean effects on cell growth were extracted. Highlighted are the nodes that give at least 2% of the maximal effect. Inhibited nodes that give the desired effect (growth reduction) are depicted in blue, and inhibited nodes with the opposite effect (growth increase) are depicted in yellow.
(EPS)

**S7 Fig. The effect on predicted apoptosis due to single node inhibition.** All individual network models were simulated under the effect of different levels of the input strength of an *in silico* inhibitor for each target present in the model. From these simulations, the mean effects on apoptosis were extracted. Highlighted are the nodes that result in at least 2% of the maximal effect. Inhibited nodes that give the desired effect (increase in apoptosis) are depicted in red, inhibited nodes with the opposite effect (reduction in apoptosis) are depicted in yellow.
(EPS)

**S8 Fig. Predicted effect of pairwise node inhibition on cell growth.** The effect on cell growth is computed for each target combination averaged over 101 network model predictions. The complete set of predictions of pairwise inhibition of molecular nodes (proteins and phosphoproteins) is displayed in the heatmap. The diagonal elements represent predictions of single target inhibition. This heatmap contains the complete data, a subset of which was included in Fig 5.
(EPS)

**S9 Fig. Predicted effect of pairwise node inhibition on apoptosis.** The effect on apoptosis is computed for each target combination averaged over 101 network model predictions. The complete set of predictions of pairwise inhibition of molecular nodes (proteins and phospho-proteins) is displayed in the heatmap. The diagonal elements represent predictions of single target inhibition. This heatmap contains the complete data, a subset of which was included in Fig 5.
(EPS)

**S10 Fig. Comparison between mean values for drug sensitivity from [28] and model-based predictions of the effect on cell growth.** The means and standard deviations per target

protein (data from S2 Table) Table are compared (left). The same mean values without error-bars (right).
(EPS)

**S11 Fig. Predicted temporal patterns of growth and apoptosis.** The mean predicted growth response (top row, blue line) and apoptosis response (bottom row, red line) as well as the standard deviation (gray area) from simulation of 101 created network models to the pairwise perturbations of EGFR-pY992/IRS1, EGFR-pY992/IRS1-pS636/639, and IRS1/IRS1-pS636/639.
(EPS)

## Acknowledgments

We thank Debbie Bemis for support, Berthold Gierke and Michael Pawlak at NMI Reutlingen for generating Dataset 2, John Ingraham for suggesting the use of the Adam optimizer in TensorFlow, as well as Emek Demir, Özgun Babur, and Augustin Luna for insightful discussions on protein–protein interactions.

## Author Contributions

**Conceptualization:** Elin Nyman, Richard R. Stein, Anil Korkut, Nicholas P. Gauthier, Chris Sander.

**Data curation:** Elin Nyman, Richard R. Stein, Xiaohong Jing, Weiqing Wang, Nicholas P. Gauthier.

**Formal analysis:** Elin Nyman, Richard R. Stein, Nicholas P. Gauthier.

**Funding acquisition:** Chris Sander.

**Investigation:** Xiaohong Jing, Weiqing Wang, Benjamin Marks.

**Methodology:** Elin Nyman, Richard R. Stein, Chris Sander.

**Resources:** Ioannis K. Zervantonakis, Chris Sander.

**Supervision:** Nicholas P. Gauthier, Chris Sander.

**Validation:** Elin Nyman, Richard R. Stein.

**Visualization:** Elin Nyman, Richard R. Stein, Nicholas P. Gauthier.

**Writing – original draft:** Elin Nyman, Richard R. Stein, Anil Korkut, Nicholas P. Gauthier, Chris Sander.

**Writing – review & editing:** Elin Nyman, Richard R. Stein, Nicholas P. Gauthier, Chris Sander.

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
