## [Decision Letter · Decision Letter 0]

27 Jan 2020

Dear Dr Stein,

Thank you very much for submitting your manuscript "Perturbation biology links temporal protein changes to drug responses in a melanoma cell line" for consideration at PLOS Computational Biology. As with all papers reviewed by the journal, your manuscript was reviewed by members of the editorial board and by several independent reviewers. The reviewers appreciated the attention to an important topic. Based on the reviews, we are likely to accept this manuscript for publication, providing that you modify the manuscript according to the review recommendations.

Sincerely,

James R. Faeder

Associate Editor

PLOS Computational Biology

Douglas Lauffenburger

Deputy Editor

PLOS Computational Biology

[LINK]

Reviewer's Responses to Questions

**Comments to the Authors:**

Reviewer #1: This is a well executed study covers a good mix of both computational and biological aspects of perturbation biology and is thus a good fit for PLoS Comp Bio. The authors introce an RPPA dataset covering the time resolved response of 126 markers to 54 drug combinations and train, validate and test a semi-mechanistic, datadriven ODE-type model on this dataset. They illustrate how the model rederives known drug-target interactions and introduce a measure of importance for individual nodes. The then go on to validate the importance of individual nodes and node combinations experimentally. The paper is well written and easy to follow, altough the some aspects of the study are not covered in sufficient detail, which I outlined below. Besides that I have noted a couple minor technical aspects that I think should be adressed.

Major Comments

1) Why did the authors decide to evaluate the model on the test set based on correlation in contrast to RSS in the previous analysis? I think looking at correlation is certainly valuable, but seems a bit inconsistent with the previous analysis. Both correlation and RSS have its pros and cons so analysing both would be valueable. Moreover it would be interesting to see the dependence of model predictions on drug/marker to complement the timepoint analysis.

2) How much is RSS biased based on these strong outliers with values < -4? Overall I am bit surprised by the number of datapoints with values smaller than -3 given that the number such datapoints in figure 2 seems to be very small.

3) How do the model predictions compare to simple null models for drug interaction such as bliss independence or highest single agent?

4) Regarding the optimization method I was wondering what the motivation was to crop interaction parameters during the optimization process. Using a gradient based method, this may lead to poor optimizer performance as it introduces discontinuities in the objective function. Why not just crop values after optimization is done? Its also unclear how derivatives are computed for cropped parameters?

5) I am not entirely sure what the authors are doing for the analysis presented in figure 5. The methods section is a bit vague on what was actually done. Were curves to compute EC50 values fit to simulation results? How were individual EC50 values extrapolated to combinations handled? I believe both methods description and results section need to be a bit more verbose about methods and rationale.

6) figure 6 needs better labels or colorcoding, it is very difficult to track what is what. Its unclear how observed maximal response values were computed (bottom/right). Is this log2 fold change on the cell growth node? Shouldnt normalized cell count be the result of cell growth and cell death? The methods section mentions something about "equivalencing", but I am not sure what this is supposed to mean and whether this is relevant to this figure.

7) Although I don't want to question it the equivalence of drugs with their target nodes in the model is nontrivial and deserves a bit more explanation/justification.

8) The level of technical documentation about the employed ode solver seems a bit shallow for a computational journal. It may be adequate to cite the recent preprint on Interpretable Machine Learning for Perturbation Biology by the authors as more detailed reference for the employed methodology.

The tensorflow optimization seems to use an explicit euler scheme without stepsize control for integration, which seems a bit unorthodox given the fact that many biological systems exhibit timescale separations which renders the underlying equations stiff. The model formulation and parameter boundaries may prevent this from happening, but it would be reassuring to validate the correctness of solutions with a state of the art ode solver with implicit integration scheme and adaptive step-size control such as scipy.integrate.solve_ivp (python) or ode15s (matlab)

9) In the discussion the authors claim. "Therefore, to avoid mis-interpretation of predictions, it is important to always study a set of obtained network models, and not only the single best solution". Although I agree with this notion, the authors don't seem to follow their own advice to closely, at least according to what was described in the paper. I would be good to know at which point uncertainty of predictions was evaluated in this study and when model averaging etc was performed. The only figure in which I could spot errorbars for multiple model realizations was in figure 3.

The authors should also note that just running multiple optimization runs does not warrant proper uncertainty analysis (c.f. "Uncertainty Analysis for Non-identifiable Dynamical Systems: Profile Likelihoods, Bootstrapping and More"). I am aware that proper uncertainty analysis using profile likelihood or bayesian methods is not realistic for models of the considered size and the authors approach is thus justified, but I think the paper should explicitely state this in the discussion.

Minor Comments:

Some supplemental figures are blurry in preview, I believe thats a matlab artefact with known workaround?

The importance of EGFR in the model highlights the importance of paracrine signaling in drug response. Accordingly, the authors may want to reconsider treating cell death/growth as terminal nodes in the model given that both may affect the degree of paracrine signaling. I don't think this needs to be adressed within the scope of this paper though.

There are still a couple of typos in the manuscript, the authors may want to recheck the manuscript.

Reviewer #2: In this manuscript titled Perturbation biology links temporal protein changes to drug responses in a melanoma cell line, Nyman et al. presented a completely data-driven approach of drug response prediction, building upon their previously developed framework of modeling the dynamic changes of cellular molecules under perturbations with a series of coupled nonlinear ordinary differential equations. They fitted the model using time-dependent RPPA data under various drug (combination) treatments in the A2058 melanoma cell line, applied the model to propose efficient novel drug treatments, and experimentally validated the proposed treatments. Overall, we find the work quite novel and interesting.

We have previously reviewed this manuscript during the authors’ submission to another journal. We are glad to see and appreciate that the authors have properly corrected some of the issues we brought up last time and largely improved the manuscript. However, we feel that some of our previous concerns have not been sufficiently addressed and therefore suggest a further revision. Here we re-discuss these issues as follows.

First, to recapitulate our previous major comments:

The authors made predictions on drug effects based on both cell growth and apoptosis, however it seems that the authors selected and tested particular drugs (and drug combinations) only based on the effect on cell growth (“normalized cell count” as in Figure 6). For completeness it is desirable to have additional validations specifically of the predicted drug effects on apoptosis (with caspase fluorescent assay). This can also be important since some perturbations (e.g. EGFR inhibition, Figure 5) were predicted to strongly suppress cell growth but only weakly trigger apoptosis. If validated it may testify the additional value of the model in revealing the context-specific mechanism of the drug action.

Most of the predictions the authors chose for validation are positive cases, i.e. where the treatment is predicted to be effective. While this can provide measures of sensitivity of the predictions, the specificity of the predictions is not well-accessed. It can be desired to test a few more cases of negative prediction to evaluate the false positive rate.

Essentially, our consideration underlying both of the above two comments is that experimentally testing the “negative cases”, although not interesting for application, is nonetheless important in thoroughly evaluating the method. Based on the limited one or two negative cases the authors have already tested, we have a concern that the method may suffer from low specificity. Moreover, testing the negative cases are not entirely biologically meaningless, e.g. in comment 1, if neutral effect on apoptosis can be validated, it will provide mechanistic insight for the action of the drug (since it does inhibit cell growth), and will help to demonstrate extra values of the authors’ method. In summary, we therefore think that it’s desirable to address at least one of the above comments. If additional biological experiments are not feasible, we think the authors may at least try to validate some of these by comparing to published data or literature and provide a proper discussion of these issues.

**Have all data underlying the figures and results presented in the manuscript been provided?**

Reviewer #1: Yes

Reviewer #2: Yes

PLOS authors have the option to publish the peer review history of their article (what does this mean?). If published, this will include your full peer review and any attached files.

Reviewer #1: No

Reviewer #2: No
---

## [Decision Letter · Decision Letter 1]

24 Apr 2020

Dear Dr Stein,

We are pleased to inform you that your manuscript 'Perturbation biology links temporal protein changes to drug responses in a melanoma cell line' has been provisionally accepted for publication in PLOS Computational Biology.

Best regards,

James R. Faeder

Associate Editor

PLOS Computational Biology

Douglas Lauffenburger

Deputy Editor

PLOS Computational Biology

Reviewer's Responses to Questions

**Comments to the Authors:**

Reviewer #1: The authors have addressed all my concerns in an adequate fashion and I recommend the acceptance for publication.

Reviewer #2: We thank the authors for addressing our comments. The manuscript is now suitable to be published.

**Have all data underlying the figures and results presented in the manuscript been provided?**

Reviewer #1: Yes

Reviewer #2: Yes

PLOS authors have the option to publish the peer review history of their article (what does this mean?). If published, this will include your full peer review and any attached files.

Reviewer #1: No

Reviewer #2: No

---

## [Editor Report · Acceptance letter]

17 Jun 2020

PCOMPBIOL-D-19-02028R1 

Perturbation biology links temporal protein changes to drug responses in a melanoma cell line

Dear Dr Stein,

I am pleased to inform you that your manuscript has been formally accepted for publication in PLOS Computational Biology. Your manuscript is now with our production department and you will be notified of the publication date in due course.

With kind regards,

Laura Mallard
